# Investigation and assessment of ecological water resources in the salt marsh area of a salt lake: A case study of West Taijinar Lake in the Qaidam Basin, China

Lu Zhao[☯], Xiao Wang[ID]*[☯], Yujun Ma[‡], Shuya Li[‡], Liuzhi Wang[‡]

School of Chemical Engineering, Qinghai University, Xining, Qinghai, China

☯ These authors contributed equally to this work.
‡ These authors also contributed equally to this work.
* wangxiao1969@163.com

**Data Availability Statement:** All relevant data are within the manuscript and its Supporting Information files.

## Abstract

The water ecology of salt marshes plays a crucial role in climate regulation, industrial production, and flood control. Due to a poor understanding of water ecology and the extensive mining of salt resources, concerns are mounting about declining groundwater levels, shrinking salt marshes, and other problems associated with the simple yet extremely fragile water ecosystem of salt marshes in arid salt lake areas. This study assessed the ecological status of water resources in the downstream salt marsh area of West Taijinar Lake in the Qaidam Basin, China (2010–2018). Using data from a field investigation, the water ecosystem was divided into an ecological pressure subsystem, an environmental quality subsystem, and a socio-economic subsystem according to an analytic hierarchy process. Each subsystem was quantitatively assessed using the ecological footprint model, the single-factor index, and available data for the salt marsh area. The results showed that water resources were always in a surplus state during the study period, whose development and utilization had a safe status. Surface water had low plankton diversity with no evidence of eutrophication, but its $Cl^-$ and $SO_4^{2-}$ concentrations were too high for direct industrial water uses. Groundwater quality was classified into class V because of high salt concentrations, which could be considered for industrial use given the demand of industrial production. The socio-economic efficiency of water resources was high, as distinguished by decreased water consumption per 10,000 yuan GDP and excellent flood resistance. In conclusion, the ecological status of water resources was deemed good in the study area and this could help sustain regional development. However, since the water ecology in this area is mainly controlled by annual precipitation, it would be challenging to deal with the uneven distribution of precipitation and flood events and to make full use of them for groundwater recharge. This study provides insight into the impact of salt lake resource exploration on water ecology, and the results can be useful for the rational utilization of water resources in salt marshes in other arid areas.

**Funding:** This research was financially supported by the Applied Basic Research Project of the Department of Science and Technology of Qinghai Province, China (Grant No.2019-ZJ-7043). The funders had no role in study design, data collection and analysis, decision to publish, or preparation of the manuscript.

**Competing interests:** The authors have declared that no competing interests exist.

# 1 Introduction

A salt marsh is a terrestrial ecosystem that is excessively wet or seasonally waterlogged, with saline soil and growing halophytes. Salt marshes are recognized worldwide for their various ecosystem services, including water quality improvement and carbon sequestration [1, 2]. They are present in a variety of environments, such as coasts, estuaries, arid or semi-arid steppes, salt lakes, and even deserts. For example, many salt marshes exist in the Bahía Blanca Estuary in South America [3]. Additionally, there are 28 salt marshes in the St. Lawrence River Estuary (including the Saguenay Fjord) in North America [4] and 757 salt marshes in the Parramatta River–Sydney Harbor in Australia [5]. In China, salt marshes are mainly distributed on the Inner Mongolia Plateau and in Xinjiang Uygur Autonomous Region in an arid or semi-arid climate, and on the Qinghai–Tibet Plateau in an alpine climate [6, 7]. In particular, the largest contiguous area of salt marshes is found in the Qaidam Basin in southwest China [8].

A salt marsh constitutes a type of ecological interface between water and land, one that provides a vital transitional ecotone for aquatic animals and plants. The salt marsh also functions as a buffer zone for natural flood control [9] and so it has beneficial effects on the local environment. In recent decades, the area of tidal and estuarine salt marshes has dramatically decreased and theirsalt marsh ecosystems have become severely degraded or destroyed due to multiple factors, such as sea level rise, agricultural runoff, and land development [10, 11]. Previous studies have mainly focused on the restoration of tidal or estuarine salt marsh ecosystems and the cultivation of salt-tolerant seedlings or on the structure and composition of biological communities [12]. Plankton communities in salt marshes are robust indicators of environmental quality, and analyzing their levels of diversity from a historical perspective could provide insight into large-scale environmental change [13, 14].

A salt marsh in arid areas differs from tidal and other types of salt marshes in terms of its higher salinity [15], which is mainly due to continuous drought, severe water shortage, and high evaporation in the arid environment. In the Qaidam Basin, the salt marsh areas of salt lakes are rich in salt resources [16], which co-exist in solid-liquid forms and are dominated by liquid ores. The industrial exploitation of these salt resources is mainly done using injection wells coupled with open-channel extraction [17] under drought and evaporation conditions. In this context, water ecology plays a key role in the exploitation of salt resources. In recent decades, ecological problems such as declining groundwater levels and the shrinking salt marshes have become increasingly prominent in the salt marsh areas of salt lakes because of an inadequate understanding of their water ecology and the extensive mining of their salt resources. This situation has shortened the service life of salt mines [18] and led to a waste of resources. Therefore, it is crucial to investigate the current status of water ecology in the salt marsh areas of salt lakes, since this could provide valuable and timely information for improving the rational use of water resources and to facilitate the development and utilization of salt lake resources.

Currently, the Qaidam Basin Salt Lake is one of the largest inland salt lakes in China and harbors much industrial value. It has the longest exploitation history and the most mature technology to extract salt lake resources in China. With the extensive exploitation of its resources, the Qaidam Basin Salt Lake is facing several increasingly prominent problems, namely a drop in groundwater levels, a reduction in salt marsh areas, and an increase in desertification. Generally, when local overexploitation and local surplus co-occur, this adversely affects the comprehensive utilization of salt lake resources and their regional coordinated development. However, if water resources in this salt marsh area could be reasonably utilized to supplement the brine in the pressure-bearing layer, the sustainable exploitation of salt lake could be effectively realized. A case in point is the West Taijinar Lake, a seasonal tail lake in

the middle of the Qaidam Basin. An investigation and assessment of ecological water resources in the salt marsh area of that lake has provided insight into the impact on water ecology from exploitation of salt lake resources. Such findings could prove useful for strengthening the rational utilization of water resources in salt marshes of those lakes and in other arid areas.

When compared with other salt marsh ecosystems, the salt marshes of salt lakes are different in terms of water ecology because their water resources are mostly supplied to industrial production. Considering the influence on it from industrial mining alone, the water ecosystem of salt marshes in salt lakes is simple yet extremely fragile. Previous research in the West Taijinar Lake has focused on its salt resources [19], while assessments of water resources in the salt marsh area of this lake are less reported. It is especially necessary to establish an appropriate assessment indicator system based on the functioning of water resources in a given salt marsh area. The analytic hierarchical process (AHP) proposed by Saaty has been widely used for the assessment of water resources in lakes, rivers, and wetlands [20, 21]. This method can be used to reliably assess the current status of water ecology in salt marshes at three criterion levels: ecological pressure, environmental quality, and social economy.

The term ecological footprint refers to the area of productive land required for the resources of productive consumption and waste disposal for a certain population. It indicates the scale of the environmental impact of the regional population and the demand for sustainable living at the current technology and consumption level [22]. Ecological footprint analysis is mainly used to assess the ecological pressure of water resources by measuring their safety, bearing capacity, and sustainable utilization [23–25]. Additionally, the single-factor index is commonly used for water quality assessment because of its simplicity and applicability [26, 27]. This method selects appropriate water quality indicators for comparison with standard values, and it evaluates water environmental quality by following the principle of 'choosing better than worse'. The socio-economic assessment of water resources is often conducted according to the ecological background of the study area.

The objective of this study was to investigate and assess the ecological status of water resources in the downstream salt marsh area of a magnesium sulfate-subtype salt lake: the West Taijinar Lake, of the Qaidam Basin. The AHP was implemented to establish an assessment indicator system with three subsystems based on a field investigation, water sampling, and laboratory analysis. The ecological footprint model and single-factor index were then used to assess each subsystem qualitatively. This study can provide useful information on water ecology in a salt marsh area. Its results could contribute to a better understanding of the impact of resource exploitation on water ecology, and how to facilitate the rational utilization of water resources in salt marshes in other arid areas.

## 2 Materials and methods: http://dx.doi.org/10.17504/protocols.io.bq3vmyn6

### 2.1 Study area

The West Taijinar Lake is located in the central part of the Qaidam Basin (Fig 1), at ~2680 m above sea level (93˚13'–93˚34' E, 37˚33'–37˚53' N). This lake's water is recharged by the Nalenggele River, whose tributaries run down from the northern slope of Kunlun Mountains [28]. Due to the influence of climate, the runoff of from the Nalenggele River mainly occurs in summer, marked by uneven distribution and dynamic change in any given year [29]. Ultimately, only a small volume of runoff flows into the downstream Taijinar River, which is divided into the East and West Taijinar Lakes. There is obvious seasonal variation in the runoff of surface rivers including the West Taijinar River and its tributary—the West Tai River that receive a mix of precipitation, surface runoff, and underground runoff. The groundwater in

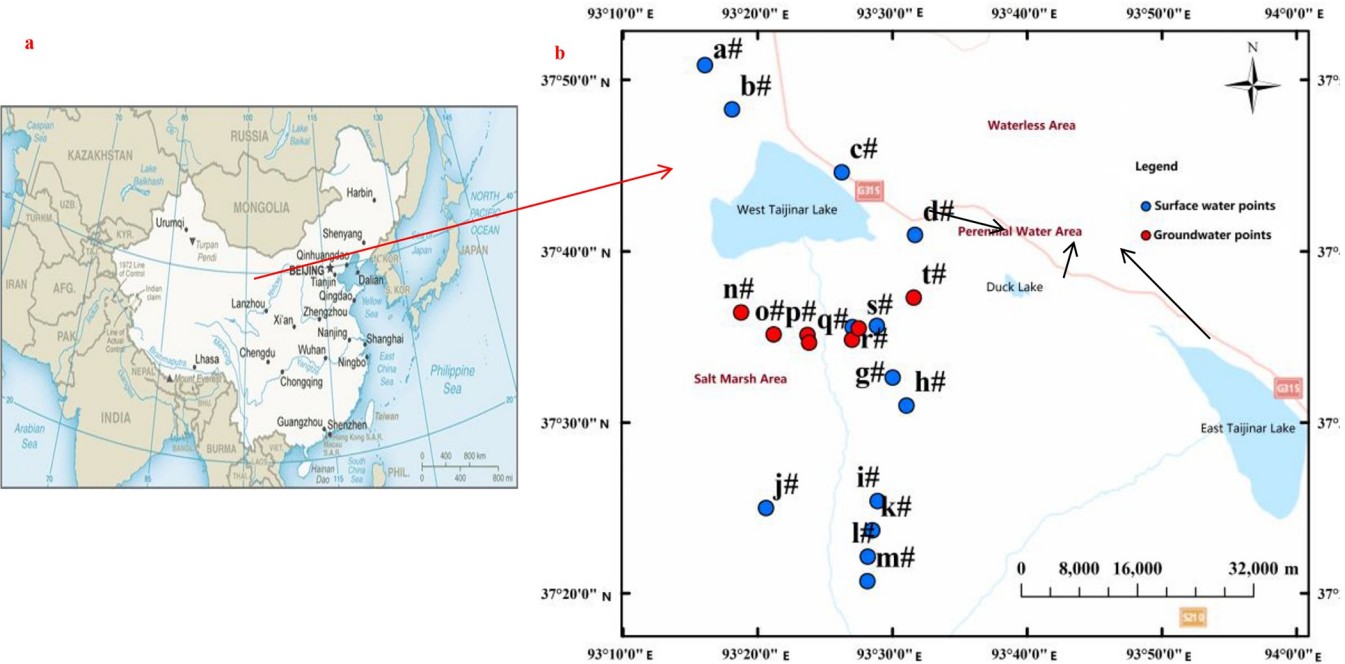

**Fig 1. Geographical location of the salt marsh area and spatial distribution of the sampling points in the West Taijinar Lake, Qaidam Basin.** a) reprinted from [CENTRAL INTELLIGENCE AGENCY] under a CC BY license, with permission from [CENTRAL INTELLIGENCE AGENCY], original copyright [2020]; b) the basemap reprinted from [ArcMap 10.2] under a CC BY license, with permission from [Esri Master License Agreement], original copyright [2019].

the salt marsh area can be divided into intercrystalline latent brine, pore latent brine, and intercrystalline pressure brine [30], and it is mainly recharged by surface water. Because rich mineral resources co-occur in solid-liquid forms [31, 32] and element contents vary at different levels, the surface water and groundwater have distinct colors ranging from colorless to light yellow.

The West Taijinar Lake is a seasonal magnesium sulfate-subtype lake downstream of the Nalenggele River, whose surroundings can be divided into a perennial water area, a waterless area, and a salt marsh area. The salt marsh area (at least 1000 km$^2$) is distributed west of National Road 315, including the area around the lake, the seasonal West Tai River, and the banks of other tributaries (Fig 1). There is almost no human activity or vegetation cover in this salt marsh area, leaving large areas of bare soil directly exposed to sunlight. Due to the influence of Late Pleistocene climate change [33], the average annual precipitation for the last three years is ~128.9 mm in the salt marsh area, while its average annual evaporation (~1200–3500 mm) is 9–28 times higher than precipitation under drought conditions [34]. The soil in the salt marsh area consists of silt, fine sand, sandy clay, and halite, mainly formed by lacustrine deposits [35]. Most of soil here is classified as desert saline soil, whose salt content is high. Severe salinization occurs in the upper soil layer, with precipitated white salt particles visible on the surface.

## 2.2 Sampling and data collection

Sampling in the study area did not require any permits because our research was supported by the institute and had been approved by Science and Technology of Qinghai Province, China. Both the sampling and field investigation were carried out in the downstream salt marsh area,

**Table 1. Geographic coordinates and elevation for the data set.**

| Types | Sampling points | Latitude | Longitude | Elevation /m | Sampling points | Latitude | Longitude | Elevation /m |
|---|---|---|---|---|---|---|---|---|
| **Surface water** | a# | 37.84758611 | 93.26803889 | 2622 | h# | 37.51648917 | 93.51747139 | 2636 |
| | b# | 37.80454889 | 93.30165194 | 2623 | i# | 37.42336111 | 93.48155556 | 2702 |
| | c# | 37.74357222 | 93.43734722 | 2688 | j# | 37.41662222 | 93.34359722 | 2718 |
| | d# | 37.68273333 | 93.52798333 | 2685 | k# | 37.39474722 | 93.47520556 | 2706 |
| | e# | 37.59298500 | 93.45051194 | 2634 | l# | 37.36905556 | 93.46951111 | 2709 |
| | f# | 37.59427500 | 9348080833 | 2692 | m# | 37.34522778 | 93.46905833 | 2712 |
| | g# | 37.54386333 | 93.50039222 | 2696 | | | | |
| **Under-groundwater** | n# | 37.60728806 | 93.31288917 | 2621 | r# | 37.58058333 | 93.45000806 | 2608 |
| | o# | 37.58588194 | 93.35307083 | 2633 | s# | 37.59173028 | 93.45828222 | 2615 |
| | p# | 37.58525139 | 93.39495083 | 2627 | t# | 37.62156583 | 93.52627694 | 2627 |
| | q# | 37.57781500 | 93.39697778 | 2626 | | | | |

within 50 km of the West Taijinar Lake, in April 2019. Random sampling of water was conducted at 5–10 km intervals in the accessible parts of the salt marsh area (Fig 1, Table 1). Thirteen surface water samples were collected from the downstream tributary of the West Taijinar River using polyethylene bottles (1000 mL), while seven underground water samples were obtained from existing boreholes using a simple water collector.

For the water quality analysis, polyethylene bottles were filled with no less than 1000 mL of water per sample. For the surface water, its pH was measured using a pH meter (FE20; METTLER TOLEDO, Zurich, Switzerland). Suspended solid (SS) was analyzed by the gravimetric method [36]. The nephelometric turbidity unit (NTU) was determined using a turbidity meter (2100Q; HACA, Loveland, USA). Total iron (Fe) was measured by inductively coupled plasma-atomic emission spectrometry (ICAP6300; Thermo Fisher Scientific, Waltham, MA, USA). Chloride ($Cl^-$) and sulfate ($SO_4^{2-}$) concentrations were respectively analyzed by silver nitrate titration [37] and volumetric method [38]. Chemical oxygen demand ($COD_{cr}$) was quantified by the dichromate method [39], while ammonia nitrogen ($NH_3$-N) concentration was determined by Nessler's reagent spectrophotometry [40]. For the groundwater, its pH, NTU, Fe, $Cl^-$, and $SO_4^2$ levels were determined as described above. Additionally, total dissolved solid (TDS) was analyzed by the weighing method. Sodium ion ($Na^+$), magnesium ion ($Mg^{2+}$), calcium ion ($Ca^{2+}$) and total boron (B) were all analyzed by inductively coupled plasma-atomic emission spectrometry.

For the plankton community analysis, 13 surface water samples (at least 500 mL) were collected with a simple water collector from the shallow (5–15 cm below the water surface), medium, or deep (5–15 cm above the river bed) depths in the downstream tributary of the West Taijinar River. Samples collected at the same point were mixed equably and concentrated randomly to give 10–20 mL per sample, followed by filtration with 25 # plankton filter and storage in sealed glass bottles with 1 mL of formol [41]. Qualitative and quantitative analyses of plankton communities were performed using an optical microscope (SAGA SG-300; Suzhou Shenying Optical Co., Ltd., Suzhou, China). The morphology of plankton was observed and identified at the genus level according to a previous study, then the number of individuals in each genuswas recorded [42]. The diversity of plankton was estimated by Shannon's diversity index (Eqs 1 and 2) [43].

$$H' = -\sum p_i \times \ln p_i \tag{1}$$

$$p_i = n/N \tag{2}$$

where, $H'$ is the Shannon's diversity index, $N$ is the total number of individuals, and $n$ is the number of individuals for species $i$.

Only herbs were collected from the salt marsh area lacking shrubs, trees, or other tall plants. Considering the extremely low density and uneven distribution of plants, the salt marsh area was divided into four units corresponding to the east, west, north, and south directions. Multi-point random sampling was conducted based on the field survey to record the number of plant species, plant height, and coverage area. Photographs were taken to identify plants at the genus level with reference to a site map [44].

The salt marsh area in the Qaidam Basin is at least 40,000 km$^2$, accounting for 1/6 of that basin's total area. Due to a lack of other data for this salt marsh area and considering its similarity to the Qaidam Basin in terms of water ecology [45], water yield and socio-economic data were obtained from the Qinghai Statistical Yearbook (Bureau of Statistics of Qinghai, 2011–2019) and the Qinghai Water Resources Bulletin (Qinghai Water Resources Bureau, 2010–2018). The total amount of water resources included precipitation, surface water, and groundwater, while water consumption included the water used for farmland irrigation, forestry, herding, fishery, livestock, industry, urban public, residential life, and the ecological environment in the Qaidam Basin during the period 2010–2018 (Table 2).

## 2.3 Assessment methods

### 2.3.1 Establishment and weight assignment of the assessment indicator system

The AHP was used to assess the ecological status of water resources in the salt marsh area. Briefly, the assessment indicator system for water resources divided them into a target layer, a criterion layer, and an indicator layer according to their characteristics such as simple water ecosystem, rich mineral resources, and primary industrial use (Fig 2).

Based on the assessment results of the indicators (C1–C8), the measurement grades of the criteria (B1–B3) were ranked into excellent, good, ordinary, and poor on the Likert scale and respectively assigned integer values 4, 3, 2, and 1. Nine discrimination grades proposed by Saaty were used to quantify the criteria subjectively. Then, the weight values ($W$), eigenvector ($\lambda$), and maximum eigenvalue ($\lambda_{max}$) of the criteria were calculated using Excel 2010 (Microsoft Corp., Redmond, WA, USA). The uniformity test was performed using Eqs 3 and 4 [46]:

$$CI = \frac{\lambda_{\max} - n}{n - 1} \qquad (3)$$

Table 2. Water yield and socio-economic data of water resources in the Qaidam Basin (2010–2018).

| Year | 2010 | 2011 | 2012 | 2013 | 2014 | 2015 | 2016 | 2017 | 2018 |
|---|---|---|---|---|---|---|---|---|---|
| Total amount /10$^8$ m$^3$ | 83.61 | 62.51 | 76.54 | 50.08 | 51.75 | 66.32 | 63.47 | 70.49 | 91.29 |
| Total consumption /10$^8$ m$^3$ | 8.1883 | 8.6142 | 6.8062 | 6.7894 | 5.9168 | 6.1886 | 6.1789 | 5.9478 | 6.3889 |
| Industrial consumption /10$^8$ m$^3$ | 3.0168 | 3.0864 | 0.8787 | 0.9406 | 0.6697 | 0.8981 | 0.8195 | 0.7984 | 0.9077 |
| Ecological consumption /10$^8$ m$^3$ | 0.4054 | 0.1339 | 0.0297 | 0.0303 | 0.1071 | 0.1268 | 0.3245 | 0.3807 | 0.4820 |
| Resident life consumption /10$^8$ m$^3$ | 0.1137 | 0.1311 | 0.0898 | 0.0943 | 0.0957 | 0.1017 | 0.1041 | 0.1208 | 0.1241 |
| Animal husbandry consumption /10$^8$ m$^3$ | 1.3519 | 1.6179 | 2.0529 | 2.1307 | 1.8928 | 2.6364 | 2.3212 | 2.3167 | 2.3855 |
| Farmland consumption /10$^8$ m$^3$ | 3.3005 | 3.6448 | 3.7551 | 3.5935 | 3.1515 | 2.4256 | 2.6096 | 2.3312 | 2.4916 |
| Calculated area /10$^2$ hm$^2$ | 257765 | 257765 | 257765 | 257765 | 257765 | 257765 | 257765 | 257765 | 257765 |
| Population / p | 385899 | 395888 | 403067 | 408200 | 412461 | 402069 | 404275 | 405658 | 404892 |

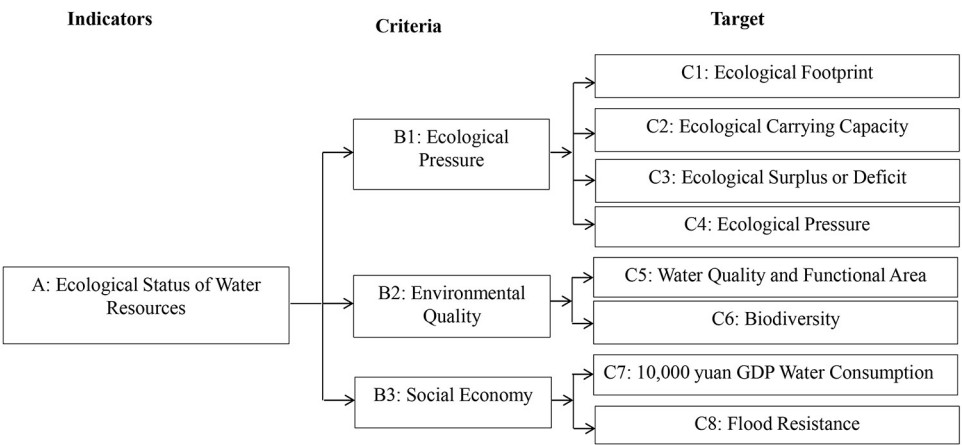

**Fig 2. The assessment indicator system of water resources based on analytical hierarchical process.**

$$CR = \frac{CI}{RI} \tag{4}$$

where *CI* is the consistency indicator; *n* is the number of criteria (*n* = 3); *CR* is the consistency ratio; and *RI* is the uniform average index (*n* = 3, *RI* = 0.52). If *CR* < 0.10, the uniformity test passed and *W* was meaningful; otherwise, *W* had to be reassigned. The value of each criterion was multiplied with their *W* to obtain the final value of the target, which was then compared with the quantitative assessment of grading standards [47, 48].

**2.3.2 Ecological pressure assessment.** To assess the ecological pressure of water resources in the salt marsh area, the ecological deficit or surplus and an ecological pressure index of water resources [49] were obtained by comparing their ecological footprint and ecological carrying capacity, using the methodology described by Wackernagel et al. [50]. For a given region, the indicators of ecological pressure were calculated using Eqs 5–8 [51]:

$$EF_W = N \times ef_w = r_w \times (W/P_w) \tag{5}$$

$$EC_W = N \times ec_w = (1 - 60\%) \times \psi_w \times r_w \times (Q/P_w) \tag{6}$$

$$EZ_W = EC_w - EF_w \tag{7}$$

$$EQ = EC_w - EF_w \tag{8}$$

where, $EF_W$ is the ecological footprint of water resources(ghm$^2$); $EC_W$ is the ecological carrying capacity of water resources (ghm$^2$); $EZ_W$ is the ecological deficit or surplus of water resources (ghm$^2$); $EQ$ is ecological pressure index of water resources; $N$ is the human population(cap); $ef_w$ is the per capita ecological footprint of water resources(ghm$^2$/cap); $ec_w$ is the per capita ecological carrying capacity of water resources(ghm$^2$/cap); $r_w$ is the global equilibrium factor for water resources ($r_w$ = 5.19, based on the calculations of WWF [52]); $\psi_w$ is the regional yield factor of land for water resources, namely the ratio of per unit yield in the study area to global production capacity of water resources, ($\psi_w$ = 0.08, based on data in Table 2); $P_w$ is the global average production capacity of water resources ($P_w$ = 3140 m$^3$/ghm$^2$ [53]); $W$ is the water consumption(m$^3$); and $Q$ is the total amount of water resources(m$^3$), of which (60% was required to maintain the ecological environment).

**Table 3. Water quality standards for industrial uses based on eight indicators.**

| Indicator | pH | SS (mg/L) | NTU | Fe (mg/L) | Cl⁻ (mg/L) | SO₄²⁻ (mg/L) | COD$_{cr}$ (mg/L) | NH₃-N (mg/L) |
|---|---|---|---|---|---|---|---|---|
| Standard | 6.5–9.0 | ≤30 | ≤5 | ≤0.3 | ≤250 | ≤600 | ≤60 | ≤10 |

SS is suspended solid. NTU is nephelometric turbidity unit. COD$_{cr}$ is chemical oxygen demand.

**2.3.3 Environmental quality assessment.** The single-factor index was used to assess the environmental quality of water resources in the salt marsh area, because the salt marsh area is rich in mineral resources and the water quality is pristine without man-made pollution [54, 55]. Currently, the water resources in this area are mainly used for industrial production processes, including cooling, washing, and supplying of products, and boiler. Here, eight representative indicators with high detection rates (Table 3) were chosen to assess the environmental quality of surface water, by following *"The reuse of urban recycling water—water quality standards for industrial uses (GBT 19923–2005)"*. Additionally, 10 representative indicators with high detection rates (Table 4) were used to assess the water quality of groundwater following the *"Standard for groundwater quality (GBT 14848–2017)"*.

**2.3.4 Socio-economic assessment.** As a decisions-making tool at economic level, socio-economic efficiency analysis uses productivity, economic and even certain social to explore the utilization efficiency of water resources [56]. The investigation and assessment of ecological water resources is a question of rationalizing the use of resources and, especially, of reducing the use of scarce and limited natural resources, or diminishing the use of other, potentially contaminating resources. In this sense, many studies have been dedicated to evaluating water use efficiency from a productive stance [57, 58], and different economic or socio-economic water use indices with agricultural, industrial, and ecosystem outputs have been proposed, such as water consumption per 10,000 yuan GDP or water consumption per capita [59]. In previous research, most evaluations of the socio-economic efficiency of water resources targeted the management of an agricultural irrigation water system. The socio-economic efficiency of a water resources system is mainly influenced by the latter's water resource use patterns and adjustment measures, which are controlled by human activities and the development of society and the economy [60]. Therefore, the selection of indicators is context-dependent and should be tailored to the actual situation of a given study area.

**Table 4. Standards for groundwater quality based on 10 indicators.**

| Indicator | Water quality class | | | | |
|---|---|---|---|---|---|
| | I | II | III | IV | V |
| pH | 6.5≤pH≤8.5 | | | 55.5≤pH<6.5 8.5<pH≤9.0 | pH<5.5 or pH>9.0 |
| NTU | ≤3 | | | ≤10 | >10 |
| TDS (mg/L) | ≤300 | ≤500 | ≤1000 | ≤2000 | >2000 |
| Na⁺ (mg/L) | ≤100 | ≤150 | ≤200 | ≤400 | >400 |
| Mg²⁺ (mg/L) | ≤10 | ≤20 | ≤50 | ≤200 | >200 |
| Ca²⁺ (mg/L) | ≤100 | ≤200 | ≤400 | ≤800 | >800 |
| B (mg/L) | ≤0.02 | ≤0.10 | ≤0.50 | ≤2.00 | >2.00 |
| Fe (mg/L) | ≤0.1 | ≤0.2 | ≤0.3 | ≤2.0 | >2.0 |
| SO₄²⁻ (mg/L) | ≤50 | ≤150 | ≤250 | ≤350 | >350 |
| Cl⁻ (mg/L) | ≤50 | ≤150 | ≤250 | ≤350 | >350 |

NTU is nephelometric turbidity unit. TDS is total dissolved solids.

In our study, industrial and ecosystem outputs mainly refer to the development and utilization of salt lakes resources. To explore the utilization efficiency of water resources, the socio-economic assessment done here was based on water consumption per 10,000 yuan GDP of industrial companies in the salt marsh area [61]. Additionally, as one of the large drainage areas in the Qaidam Basin, the West Taijinar Lake is an important flood storage and detention area, yet also threatened by extreme flooding, which has certain effects upon local industrial production. The lake's flood resistance was comprehensively assessed based on data for its slope gradient, bank height, and soil type that were collected from the salt marsh area through a field investigation [62, 63].

## 3 Investigation of ecological water resources

The investigation of ecological water resources provides raw data for their qualitative assessment. The water ecosystem in the salt marsh area was divided into an ecological pressure subsystem, an environmental quality subsystem, and a socio-economic subsystem, which were interdependent and mutually restrictive (Fig 3). Investigating the background ecological status of water resources in each subsystem could facilitate the establishment of an assessment system and the selection of assessment indicators.

### 3.1 Ecological pressure subsystem of water resources

The ecological pressure subsystem of water resources indicates whether the utilization of water resources is sustainable for a given area, which is mainly related to the human population density, total amount and consumption of water resources, and land use area. Generally, the ecological pressure subsystem of water resources in our study area is similar to that of the Qaidam Basin. The gross area of the basin is ~0.25 million km$^2$, where the population size remains low (no more than 41 million by the end of 2018; Table 2). Only two industrial parks—Qinghai CITIC Guoan Lithium Development Co., Ltd. and Qinghai Hengxinrong Lithium Technology Co., Ltd.—are located within 50 km of the salt marsh area, where the total population is always never exceeds 1000 excluding residential zones.

The total amount of water resources in the subsystem includes three components: inflow, outflow, and internal exchange. There are two pathways of water inflow into the salt marsh area: natural runoff and precipitation, both featuring dramatic seasonal change. Natural runoff

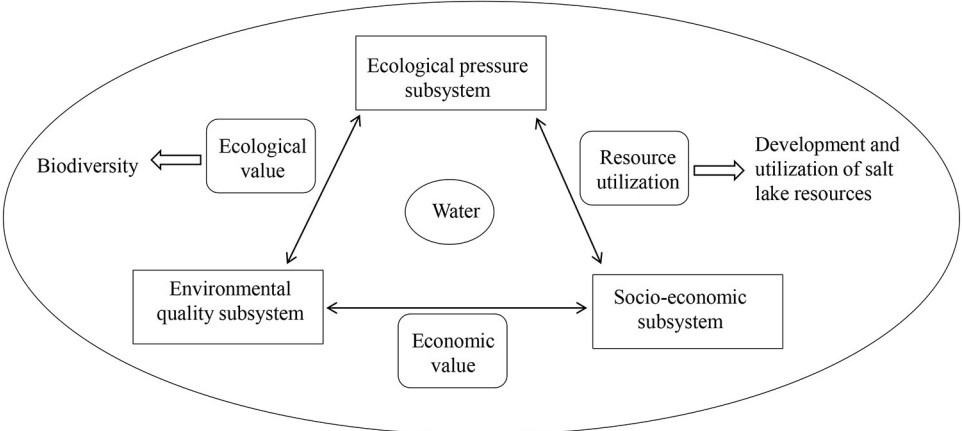

**Fig 3. The water ecosystem comprising three subsystems in the study area.**

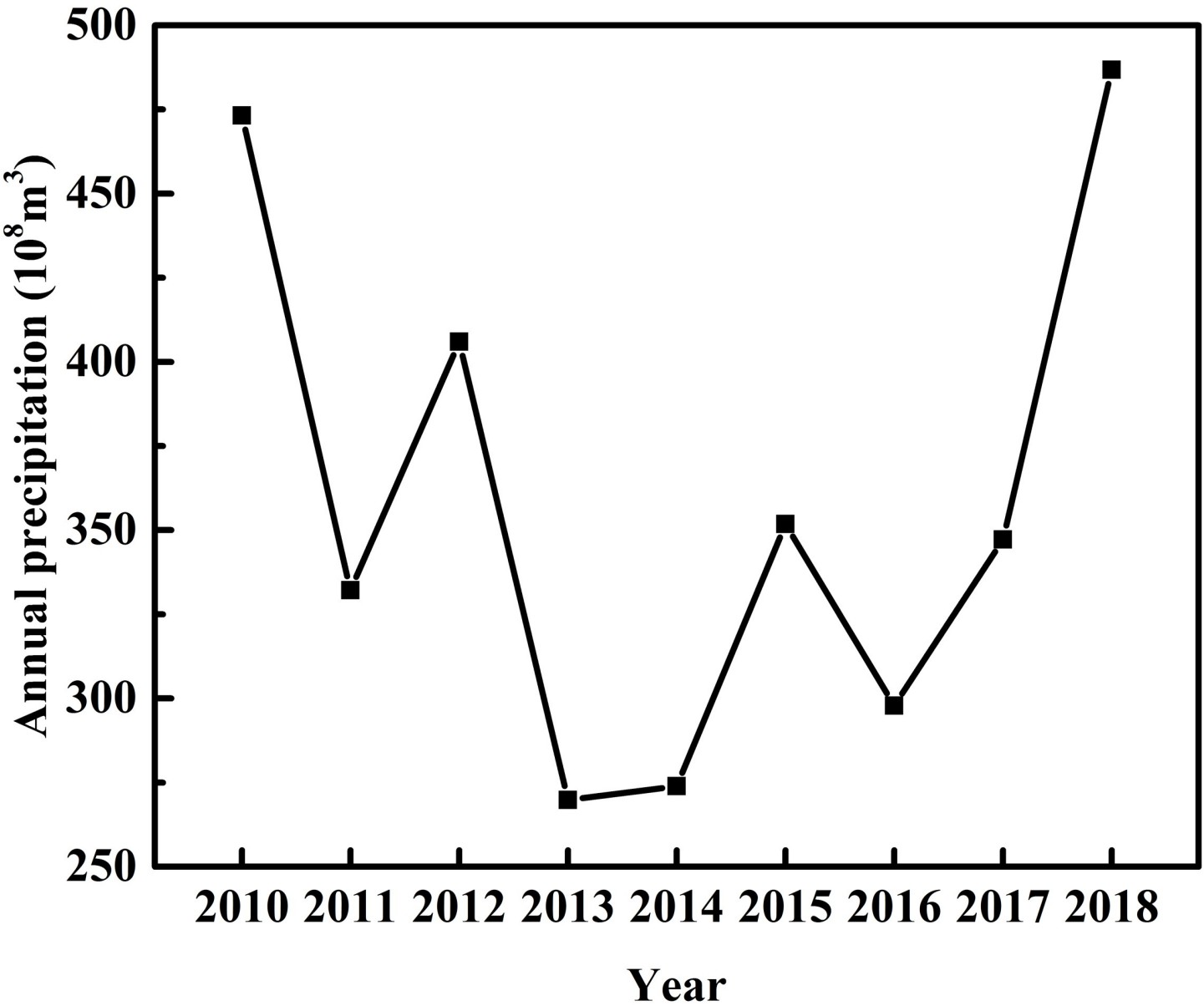

**Fig 4. Annual precipitation in the Qaidam Basin (2010–2018).**

increases with the runoff of the Nalengele River and its tributaries during the wet season (April–September), when floods may occur occasionally [64]. Moreover, natural runoff increases with the melting of ice and snow due to the influence of global warming. Precipitation is the most important source of water in the Qaidam Basin, and it directly determines the amount of water resources and the level of groundwater in the salt marsh area. Based on available data through 2018, the average annual precipitation in the basin is 35.98 billion $m^3$; but since 2013 and 2014 (both drought years), the annual precipitation has increased continuously and reached 48.68 billion $m^3$ in 2018 (Fig 4).

The water resources flowing out of the salt marsh area are mainly consumed for the production of salt lake resources, at a rate of ~6000 $m^3$/d for the two companies during the

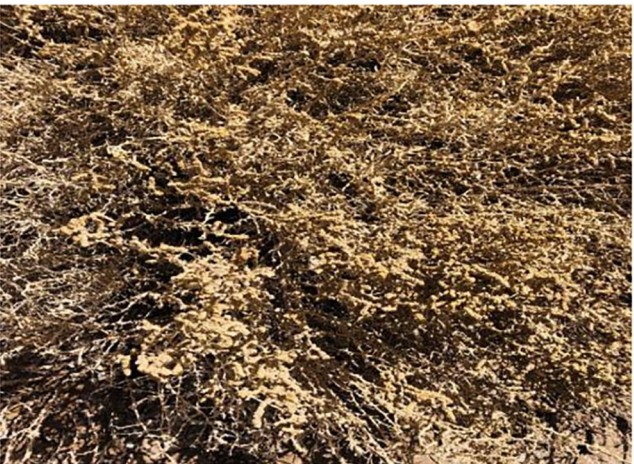

**Fig 5. A photograph of halophytic plants (*Kalidium*) growing in the downstream of West Tai River (taken on 24 April, 2019 around i# sampling point).**

manufacturing process. Currently, there is an increasing trend in the daily consumption of water with a continuous drop in the groundwater level and increasing difficulty of salt lake exploitation. Internal exchange consists of evaporation and infiltration. Because there is almost no vegetation coverage, fast evaporation occurs under strong sunlight in the salt marsh area.

## 3.2 Environmental quality subsystem of water resources

Both the ecological environment and industrial production conditions are affected by water quality. In the salt marsh area of the West Taijinar Lake, changes in water quality mainly arise from natural factors since domestic or industrial sewage is absent. The water quality is altered dynamically across years and seasons, as evinced by varied colors ranging from colorless to light yellow due to different salt ion contents. Based on our field investigation, the vegetation coverage in the study area is less than 0.1% and no animals could be found. Only small numbers of *Kalidium* plants grow downstream of the West Tai River (Fig 5), while trace amounts of algae are present in the low-lying area upstream of the river. Overall, there are no signs of eutrophication in the river water.

## 3.3 Socio-economic subsystem of water resources

The socio-economic subsystem of water resources is highly dependent on the other two subsystems that provide water for industrial production and socio-economic benefits. Industrial production in the salt marsh area mainly relies on the production capacity of salt lake mineral resources. In 2018, Qinghai CITIC Guoan Lithium Industry Development Co., Ltd. had an annual output of 2.7 million tons of potassium sulfate and 4500 tons of lithium carbonate, with ~1.1 billion yuan of business income, while Qinghai Hengxinrong Lithium Technology Co., Ltd. produced 20 thousand tons of battery-grade lithium carbonate. According to the field investigation, a large amount of surface water is discharged into the salt marsh area during the wet season (April–September), causing floods in some cases. If no flood prevention measures get implemented, this excess water would flow into salt pans, negatively affecting the operation of the industrial area and reducing the economic production capacity.

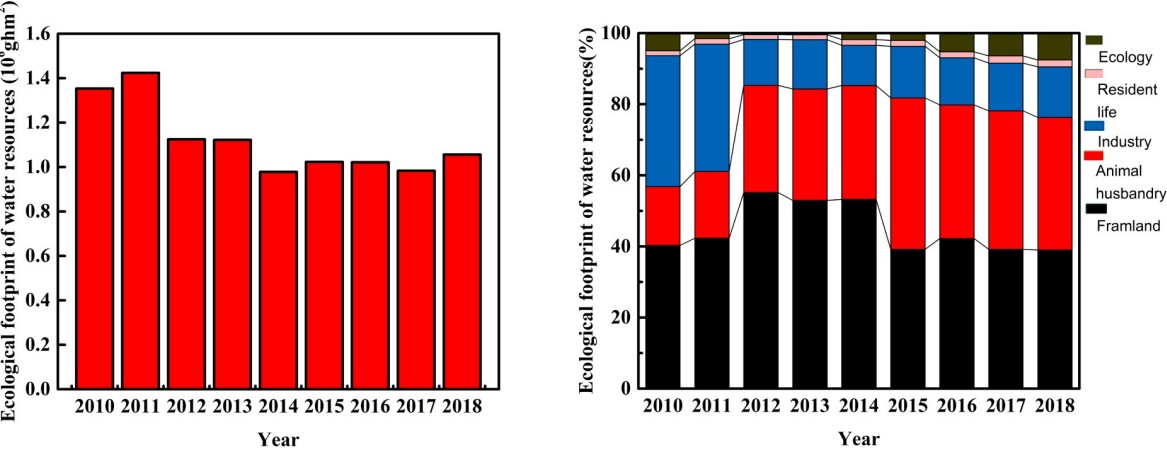

**Fig 6. Ecological footprint of water resources (left) and the proportion of different water consumption (right) in the study area.**

# 4 Assessment of ecological water resources

## 4.1 Ecological pressure of water resources

The ecological footprint, carrying capacity, and surplus of water resources in the study area are shown in Figs 6 and 7. During the study period, the ecological footprint of water resources first decreased (2010–2013) and then leveled off (2014–2018). The highest ecological footprint was $1.4238 \times 10^6$ ghm$^2$, in 2011, and the annual average was $1.1206 \times 10^6$ ghm$^2$ for the years 2010–2018 (Fig 6, left). Water resources in the study area were mainly consumed by industry and the ecological environment. However, with proposal of "green development", the local government has paid increasing attention to ecological protection and resource conservation

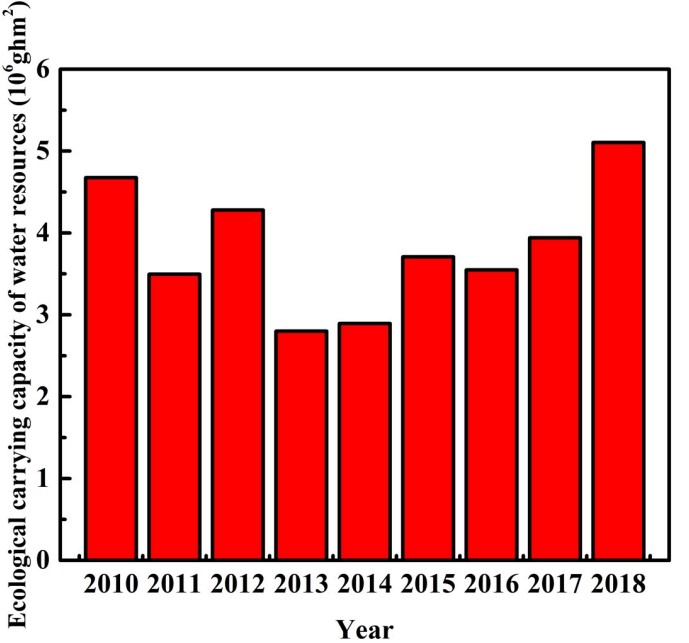

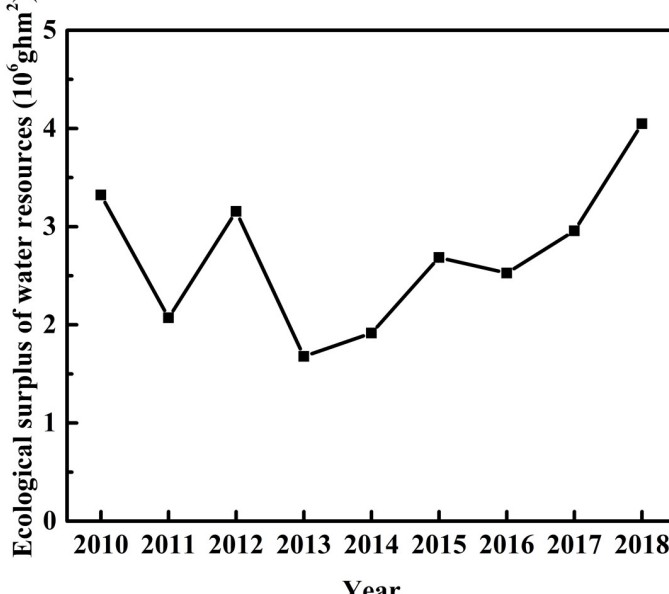

**Fig 7. Ecological carrying capacity (left) and surplus (right) of water resources in the study area.**

**Table 5. The ecological pressure index of water resources.**

| Year | 2010 | 2011 | 2012 | 2013 | 2014 | 2015 | 2016 | 2017 | 2018 |
|---|---|---|---|---|---|---|---|---|---|
| index value | 0.0290 | 0.0407 | 0.0263 | 0.0401 | 0.0338 | 0.0276 | 0.0288 | 0.0250 | 0.0207 |

in this area, and the rate of water recycling has increased [65]. Therefore, its proportion of the ecological water footprint increased gradually from 0.57% to 7.54%, while those of industrial water decreased with time. Over the years, the population density remained low in the study area (Table 2), so that the residential water footprint was generally stabled, accounting for ~1.5% of the ecological footprint (Fig 6, right).

Generally, the ecological carrying capacity of water resources initially decreased and then increased over the study period, except for an abnormal value in 2012 (Fig 7). The lowest value appeared in 2013 ($2.8000 \times 10^7$ ghm$^2$), likely related to the lowest annual precipitation in that year (Fig 4). After that, the ecological carrying capacity continuously increased, so that it was 82.3% higher in 2018. The ecological carrying capacity of water resources exhibited a trend similar to that of annual precipitation (Fig 4). This result suggests the ecological carrying capacity is positively controlled by annual precipitation in the study area. Because the distribution of annual precipitation is uneven in time and space, it nonetheless remains challenging to make good use of seasonal precipitation and flood resources in the salt marsh for groundwater recharge. The ecological carrying capacity of water resources exceeded the ecological footprint from 2010 to 2018, amounting to an ecological surplus of more than $1.6 \times 10^6$ ghm$^2$ (Fig 7). This result indicates that the water resources in the study area are in a sustainable state. The trend in the ecological surplus of water resources was consistent with that of the ecological carrying capacity, suggesting that the overall utilization rate of water resources was sufficient to maintain regional development. The ecological pressure index of water resources remained at less than 0.5, varying between 0.0407 and 0.0207 during 2010–2018 (Table 5). This result indicated that the development and utilization of water resources in the study area is in a safe state relative to the index range [66] and that its ecological pressure subsystem is good (B1 = 3).

## 4.2 Environmental quality of water resources

**4.2.1 Surface water quality.**   The genus and quantities of plankton in 13 surface water samples are summarized in Table 6. Dinoflagellates and diatoms accounted for 35% and 31% of all phytoplankton, respectively, while copepods and rotifers accounted for 43% and 42% of zooplankton, respectively. Based on these results, the phytoplankton biodiversity index ($H'_1$) and the zooplankton biodiversity index ($H'_2$) were obtained:

$$H'1 = -[0.35 \times \ln(0.35) \times \ln(0.31) + 0.21 \times \ln(0.21) + 0.13 \times \ln(0.13)]$$
$$= 1.32$$

$$H'2 = -[0.15 \times \ln(0.15) + 0.42) \times \ln(0.42) + 0.43 \times \ln(0.43)]$$
$$= 1.01$$

The biodiversity index values of phytoplankton and zooplankton were extremely low, at 1.32 and 1.01, respectively. This low plankton diversity in the salt marsh area might be related to the low temperature (~3.56°C annual mean temperature) and high salinity (~331.5 g/L) of its saline water [67].

**Table 6. Species and quantity of plankton in thirteen surface water samples from the salt marsh area.**

| Plankton | Phytoplankton | | | | | Zooplankton | | | |
|---|---|---|---|---|---|---|---|---|---|
| | Dinoflagellate | Diatom | Chlorella | Cyanobacteria | Total | Protozoa | Rotifer | Copepod | Total |
| Total number | 393 | 349 | 237 | 151 | 1130 | 40 | 112 | 115 | 267 |
| Average number per sample | 30 | 27 | 18 | 12 | 87 | 3 | 9 | 9 | 21 |
| Ratio | 0.35 | 0.31 | 0.21 | 0.13 | — | 0.15 | 0.42 | 0.43 | — |

**Table 7. Environmental quality of 13 surface water samples from the salt marsh area for industrial uses.**

| Indicator | | pH | SS | NTU | Fe | Cl$^-$ | SO$_4^{2-}$ | COD$_{cr}$ | NH$_3$-N |
|---|---|---|---|---|---|---|---|---|---|
| Usable or not | Yes | 13 | 0 | 12 | 13 | 0 | 4 | 13 | 13 |
| | No | 0 | 13 | 1 | 0 | 13 | 9 | 0 | 0 |

SSis suspended solid. NTU is nephelometric turbidity unit. COD$_{cr}$ is chemical oxygen demand.

To assess the surface water quality for industrial uses, the measured values of the eight quality indicators were compared with their standard values (Table 3). The over-standard rates of SS and Cl$^-$ were 100%, while those of SO$_4^{2-}$ and NTU respectively were 69.2% and 7.7%. The proportions of SS, Cl$^-$, SO$_4^{2-}$, and NTU in the over-standard indicators were 36.1%, 36.1%, 25.0%, and 2.8%, respectively (Table 7). These results suggest that the surface water in the study area could be directly used as industrial water. In summary, the surface water in the salt marsh area of the West Taijinar Lake exhibits a self-purification ability, with no eutrophication present; however, it fails to meet the industrial water standards due to its high levels of SS, Cl$^-$, SO$_4^{2-}$, and NTU, which could be related to salt migration driven by the interaction between surface water and saline soil precipitation events and runoff flow processes. Although the surface water is not suitable for direct industrial use, it could be considered for recharging the groundwater in the mining area.

**4.2.2 Groundwater quality.** Based on the 10 indicators of groundwater quality, their measured values were compared with the standard values (Table 4) to assess the water quality in the salt marsh area. The chemical evolution of groundwater is mainly controlled by water–rock interactions and the evaporation–crystallization process [68], and it exists in the form of saline water (TDS >2000 mg/L). Based on its pH, Fe, and NTU levels, the groundwater in the study area met the water quality standards for class I (Table 8). Specifically, the pH levels of all groundwater samples met the class I standard, while the Ca$^{2+}$ concentrations in 28.6% of groundwater samples complied with the class III standard. For the class IV, the NTU, Ca$^{2+}$, and Fe levels in 42.3%, 14.3%, and 25% of the samples met the standard, respectively. Generally, TDS, Na$^+$, Mg$^{2+}$, B, SO$_4^{2-}$, and Cl$^-$ were the main factors for the deterioration of groundwater quality, which met the class V standard (100%). In summary, the groundwater in the salt marsh area of West Taijinar Lake was affected by salt deposits and classified into class V. Due to high salt contents, the groundwater is not suitable for use drinking water, but it ould be considered for industrial uses based on the industrial water demand. In short, the environmental quality of water resources was poor (B2 = 1).

## 4.3 Social economy of water resources

From 2015 onward, the total industrial water consumption declined and total industrial output value rose yearly following the adjustments made to the production technology of

**Table 8. Environmental quality of seven groundwater samples from the salt marsh area.**

| Indicator | Total number of data | Number of samples at different water quality class | | | | |
|---|---|---|---|---|---|---|
| | | I | II | III | IV | V |
| pH | 7 | 7 | 0 | 0 | 0 | 0 |
| NTU | 7 | 2 | 0 | 0 | 3 | 2 |
| TDS | 7 | 0 | 0 | 0 | 0 | 7 |
| $Na^+$ | 7 | 0 | 0 | 0 | 0 | 7 |
| $Mg^{2+}$ | 7 | 0 | 0 | 0 | 0 | 7 |
| $Ca^{2+}$ | 7 | 0 | 0 | 2 | 1 | 4 |
| B | 7 | 0 | 0 | 0 | 0 | 7 |
| Fe | 7 | 6 | 0 | 0 | 2 | 0 |
| $SO_4^{2-}$ | 7 | 0 | 0 | 0 | 0 | 7 |
| $Cl^-$ | 7 | 0 | 0 | 0 | 0 | 7 |

NTU is nephelometric turbidity unit. TDS is total dissolved solids.

Qinghai CITIC Guoan Lithium Development Co., Ltd. and Qinghai Hengxinrong Lithium Technology Co., Ltd. Consequently, the water consumption per 10,000 yuan GDP decreased on a yearly basis. This result is indicative of an increasing trend in the utilization efficiency of water resources over time, which is beneficial for the social economy of water resources in the salt marsh area. According to the "*Code for design of embankment engineering (GB50286-2013)*", both industrial companies in the study area are large-scale enterprises, with a level-2 dam capacity for flood control, an extra height for safety at ≥0.8 m, a preliminary estimate of wave climbing at ~0.6–0.7 m, and a total superelevation of ≥1.4 m. The slope gradient of the study area ranged from 20° to 75°, and the bank height was 2–6 m. Based on the field investigation, the soil type was mainly clay soil mixed with magnesia (MgO)—a component of industrial wastes released the salt marsh area while sandy soil comprised less than 10% of the soil composition. Presently, the flood control dam is still expanding and being reinforced to provide excellent flood resistance. Based on our investigation results, the socio-economic status of water resources in the study area was in a good condition during the study period (B3 = 3).

## 4.4 Ecological status of water resources

Quantitative results of the AHP are summarized in Tables 9 and 10. The *W* was meaningful (0.05 < CR < 0.1) and the final value of the target was obtained:

$$A = 3 \times 0.524680631 + 1 \times 0.141550483 + 3 \times 0.333768887 = 2.716899$$

The result indicates that the ecological status of water resources was good in the downstream salt marsh area of the West Taijinar Lake (2.5 < A ≤ 3.5).

**Table 9. Judgment matrix of the criteria for the assessment of water resources.**

| B | B1 | B2 | B3 |
|---|---|---|---|
| B1 | 1 | 3 | 1 |
| B2 | 1/3 | 1 | 1/3 |
| B3 | 1 | 3 | 1 |

**Table 10. Quantitative results of the AHP for the assessment of water resources.**

| Criteria | W | λ | $\lambda_{max}$ | CI | CR |
|---|---|---|---|---|---|
| B1 | 0.524680631 | 1.616869852 | 3.053749835 | 0.026874918 | 0.051682534 |
| B2 | 0.141550483 | 0.427671707 | | | |
| B3 | 0.333768887 | 1.02076065 | | | |

## 5 Conclusions

Based on field investigation, this study established an assessment indicator system for the ecologic status of water resources in the downstream salt marsh area of the West Taijinar Lake. The proposed system comprised one target layer, three criterion layers, and eight indicator layers. From 2010 to 2018, the water resources in the study area had a stable surplus and the overall utilization rate was not high. Generally, the development and utilization of water resources was in a safe state, with low ecological pressure. Despite low plankton diversity and no eutrophication, the surface water did not meet the standards for industrial water uses, while groundwater resources could be considered for industrial uses based on the demand of industrial production. Considering the social economy of water resources, the water consumption per 10,000 yuan GDP declined on a yearly basis and the flood resistance was excellent. In conclusion, the water resources in the salt marsh area were in a good ecological state, which could maintain sustainable development on a regional scale. However, the water ecology in the study area was mainly controlled by annual precipitation.

The findings of this study must take into account some limitations. Firstly, due to time constraints, the surface water and groundwater throughout the Qaidam Basin could not be explored. Secondly, the proposed assessment indicator system for the ecological status of water resources is suitable for salt marshes in arid areas only; however, the index of the target layer can be added or subtracted according to the actual situation of a study area. Finding a way to accommodate and harness the uneven distribution of precipitation and flood could enable their full use for groundwater recharge, but this would be a major challenge to overcome in resolving the problems associated with falling groundwater levels and shrinking salt marshes in arid areas. Based on the investigation and assessment of ecological water resources for the whole basin, it is imperative that future research avenues focus on the sustainable allocation of water resources, namely from rainfall and floods, in the Qaidam Basin.

## Acknowledgments

We warmly thank Transcend Envirotech Consulting for the copyediting of the language usage, spelling, and grammar.

## Author Contributions

**Conceptualization:** Lu Zhao, Xiao Wang.

**Data curation:** Lu Zhao, Yujun Ma, Shuya Li, Liuzhi Wang.

**Formal analysis:** Lu Zhao, Yujun Ma.

**Investigation:** Lu Zhao, Yujun Ma, Shuya Li.

**Methodology:** Lu Zhao, Liuzhi Wang.

**Resources:** Xiao Wang.

**Software:** Lu Zhao.

**Validation:** Shuya Li.

**Writing – original draft:** Lu Zhao.

**Writing – review & editing:** Lu Zhao.

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
