## [Decision Letter · Decision Letter 0]

26 Nov 2020

PONE-D-20-31417

Investigation and assessment of ecological water resources in the salt marsh area of the salt lake—A case study of the West Taijinar Lake in Qaidam Basin, China

PLOS ONE

Dear Dr. Wang,

Thank you for submitting your manuscript to PLOS ONE. After careful consideration, we feel that it has merit but does not fully meet PLOS ONE’s publication criteria as it currently stands. Therefore, we invite you to submit a revised version of the manuscript that addresses the points raised during the review process.

We look forward to receiving your revised manuscript.

Kind regards,

Ali Kharrazi

Academic Editor

PLOS ONE

Journal Requirements:

2. In your Methods section, please provide additional location information of the sampling sites, including geographic coordinates for the data set if available.

3. In your Methods section, please provide additional information regarding the permits you obtained for the work. Please ensure you have included the full name of the authority that approved the sampling sites access and, if no permits were required, a brief statement explaining why.

4.We suggest you thoroughly copyedit your manuscript for language usage, spelling, and grammar. If you do not know anyone who can help you do this, you may wish to consider employing a professional scientific editing service.  

7. We note that [Figure(s) 1] in your submission contain map images which may be copyrighted. All PLOS content is published under the Creative Commons Attribution License (CC BY 4.0), which means that the manuscript, images, and Supporting Information files will be freely available online, and any third party is permitted to access, download, copy, distribute, and use these materials in any way, even commercially, with proper attribution. For these reasons, we cannot publish previously copyrighted maps or satellite images created using proprietary data, such as Google software (Google Maps, Street View, and Earth). For more information, see our copyright guidelines: http://journals.plos.org/plosone/s/licenses-and-copyright.

1.    You may seek permission from the original copyright holder of Figure(s) [1] to publish the content specifically under the CC BY 4.0 license. 

Reviewers' comments:

Reviewer's Responses to Questions

**Comments to the Author**

1. Is the manuscript technically sound, and do the data support the conclusions?

Reviewer #1: Yes

2. Has the statistical analysis been performed appropriately and rigorously? 

Reviewer #1: I Don't Know

3. Have the authors made all data underlying the findings in their manuscript fully available?

Reviewer #1: Yes

4. Is the manuscript presented in an intelligible fashion and written in standard English?

Reviewer #1: Yes

5. Review Comments to the Author

Reviewer #1: - the paper is well written and developed

- the concept of socio-economic efficiency is not very clear from the abstract, please explain

- socio-economic efficiency is not well discussed within the paper, please expand on this concept, what is being measured, what are the limitations of this concept and where has it been used before?

- the motivation for this study area is not clear, I recommend the authors to emphasize why did the authors decide to examine this study area

- I recommend authors to discuss the limitations found in the study

- I recommend authors to discuss future research avenues

- please improve the language and flow of the paper for minor errors and typos

6. PLOS authors have the option to publish the peer review history of their article (what does this mean?). If published, this will include your full peer review and any attached files.

Reviewer #1: No

---

## [Author Response · Author response to Decision Letter 0]

8 Jan 2021

Ms. Ref. No.: PONE-D-20-31417

Title: Investigation and assessment of ecological water resources in the salt marsh area of the salt lake—A case study of the West Taijinar Lake in Qaidam Basin, China

Journal: PLOS ONE

Dear Editor:

We have carefully considered the comments from the reviewers, and have revised the manuscript accordingly. We appreciate the editor and reviewers for the positive and constructive comments and suggestions on our manuscript. The inclusion of reviewers’ comments has improved the quality of our work.

We have highlighted all changes in the text in red color in my manuscript labeled “Revised Manuscript with Track Changes”. A point-by-point response to the editor/reviewers’ comments is given below, including line and page numbers in the revised manuscript. The line and page numbers refer to the numbers in my manuscript labeled “Revised Manuscript with Track Changes”. Please feel free to contact us if you need any further information.

Sincerely yours,

Xiao Wang 

Department: Chemical Engineering of Qinghai University

Address: Qinghai University No. 251 Ningda Road Chengbei District Xining City of Qinghai Province in China, 810016

Tel.: 86-15709781855; E-mail: wangxiao1969@163.com

 

Response to Comments

Journal Requirements:

1. Response: We would not like to make changes to our financial disclosure.

Response: We have deposited our laboratory protocols in protocols.io and have added DOI in materials and methods in page 6, line 143. 

DOI is http://dx.doi.org/10.17504/protocols.io.bq3vmyn6

Response: We have modified the manuscript with reference to PLOS ONE's style requirements, including those for file naming. Also, we have made all data underlying the findings in our manuscript fully available and the statistical analysis has been performed appropriately and rigorously. Our figures have verified using PACE.

Track Changes is as follows:

1. Title page

2. Figures and Tables

3. Language Usage, Spelling, and Grammar

4. References

5. Styles, including those for file naming

6. The contents proposed by the reviewer that needs to be revised

7. Acknowledgements

4. In your Methods section, please provide additional location information of the sampling sites, including geographic coordinates for the data set if available.

Response: Additional location information of the sampling sites, including geographic coordinates and elevation for the data set has been added in Table 1 in page 8, line 200.

Table 1. Geographic coordinates and elevation for the data set

Types Sampling points Latitude Longitude Elevation

/m Sampling points Latitude Longitude Elevation

/m

Surface water a# 37.84758611 93.26803889 2622 h# 37.51648917 93.51747139 2636

 b# 37.80454889 93.30165194 2623 i# 37.42336111 93.48155556 2702

 c# 37.74357222 93.43734722 2688 j# 37.41662222 93.34359722 2718

 d# 37.68273333 93.52798333 2685 k# 37.39474722 93.47520556 2706

 e# 37.59298500 93.45051194 2634 l# 37.36905556 93.46951111 2709

 f# 37.59427500 9348080833 2692 m# 37.34522778 93.46905833 2712

 g# 37.54386333 93.50039222 2696 

Under-groundwater n# 37.60728806 93.31288917 2621 r# 37.58058333 93.45000806 2608

 o# 37.58588194 93.35307083 2633 s# 37.59173028 93.45828222 2615

 p# 37.58525139 93.39495083 2627 t# 37.62156583 93.52627694 2627

 q# 37.57781500 93.39697778 2626 

5. In your Methods section, please provide additional information regarding the permits you obtained for the work. Please ensure you have included the full name of the authority that approved the sampling sites access and, if no permits were required, a brief statement explaining why.

Response: Sampling does not need permits, because the research supported by the institute has been approved by Science and Technology of Qinghai Province, China. So a brief statement has been added in page 7, line 178-179.

The added statement is as follows: 

“Sampling in the study area did not require any permits because our research was supported by the institute and had been approved by Science and Technology of Qinghai Province, China.”

6. We suggest you thoroughly copyedit your manuscript for language usage, spelling, and grammar. If you do not know anyone who can help you do this, you may wish to consider employing a professional scientific editing service. 

The name of the colleague or the details of the professional service that edited your manuscript

A copy of your manuscript showing your changes by either highlighting them or using track changes (uploaded as a *supporting information* file)

A clean copy of the edited manuscript (uploaded as the new *manuscript* file)

Response: We have thoroughly copyedited our manuscript for language usage, spelling, and grammar with the help of Transcend Envirotech Consulting. There is a proof in the supporting information named Editing Certificate.

7. We note that you have included the phrase “data not shown” in your manuscript. Unfortunately, this does not meet our data sharing requirements. PLOS does not permit references to inaccessible data. We require that authors provide all relevant data within the paper, Supporting Information files, or in an acceptable, public repository. Please add a citation to support this phrase or upload the data that corresponds with these findings to a stable repository (such as Figshare or Dryad) and provide and URLs, DOIs, or accession numbers that may be used to access these data. Or, if the data are not a core part of the research being presented in your study, we ask that you remove the phrase that refers to these data.

Response: In manuscript, “data not shown” are in page 17, line 409 and page 21, line 467. 

For the first one, “data not shown” has been deleted and the ecological pressure index of water resources calculated according to Eq. 8 has been shown in Table 5.

Table 5 The ecological pressure index of water resources 

Year 2010 2011 2012 2013 2014 2015 2016 2017 2018

index value 0.0290 0.0407 0.0263 0.0401 0.0338 0.0276 0.0288 0.0250 0.0207

For the latter, it does not matter for “data not shown” deletion and “data not shown” has been deleted. The trend of the water consumption per 10,000 yuan GDP of Qinghai CITIC Guoan Lithium Development Co., Ltd. and Qinghai Hengxinrong Lithium Technology Co., Ltd. were derived from the trend of total industrial water consumption and total industrial output value.

8. PLOS requires an ORCID iD for the corresponding author in Editorial Manager on papers submitted after December 6th, 2016. Please ensure that you have an ORCID iD and that it is validated in Editorial Manager. To do this, go to ‘Update my Information’ (in the upper left-hand corner of the main menu), and click on the Fetch/Validate link next to the ORCID field. This will take you to the ORCID site and allow you to create a new iD or authenticate a pre-existing iD in Editorial Manager. Please see the following video for instructions on linking an ORCID iD to your Editorial Manager account: https://www.youtube.com/watch?v=_xcclfuvtxQ

Response: An ORCID iD has been validated in Editorial Manager. The ORCID iD registered to my address wangxiao1969@163.com is https://orcid.org/0000-0002-7883-5409.

9. We note that [Figure(s) 1] in your submission contain map images which may be copyrighted. All PLOS content is published under the Creative Commons Attribution License (CC BY 4.0), which means that the manuscript, images, and Supporting Information files will be freely available online, and any third party is permitted to access, download, copy, distribute, and use these materials in any way, even commercially, with proper attribution. For these reasons, we cannot publish previously copyrighted maps or satellite images created using proprietary data, such as Google software (Google Maps, Street View, and Earth). For more information, see our copyright guidelines: http://journals.plos.org/plosone/s/licenses-and-copyright.

1. You may seek permission from the original copyright holder of Figure(s) [1] to publish the content specifically under the CC BY 4.0 license. 

Response: Figure 1 has been replaced and modified. The figure caption was changed to “Fig 1. Geographical location of the salt marsh area and spatial distribution of the sampling points in the West Taijinar Lake, Qaidam Basin, a) reprinted from [CENTRAL INTELLIGENCE AGENCY] under a CC BY license, with permission from [CENTRAL INTELLIGENCE AGENCY], original copyright [2020]; b) the basemap reprinted from [ArcMap 10.2] under a CC BY license, with permission from [Esri Master License Agreement], original copyright [2019]” in page 6-7, line 159-163.

The a) has been replaced with a map downloaded from Maps at the CIA (public domain) (https://www.cia.gov/library/publications/the-world-factbook/index.html). 

The b) was completed with ArcGIS 10.2 software, and the basemap was directly added using the ArcGIS Online function in ArcMap 10.2. The latitude and longitude, sampling points, wind rose, scaleplate, and geographic names were all added and perfected by the authors. ArcGIS 10.2 software is licensed under the Esri Master License Agreement. 

The introduction of the software includes “The map was developed by National Geographic and Esri and reflects the distinctive National Geographic cartographic style in a multi-scale reference map of the world. The map was authored using data from a variety of leading data providers, including Garmin, HERE, UNEP-WCMC, NASA, ESA, USGS, and others.” More detailed contents can be viewed through the website: https://www.arcgis.com/home/item.html?id=b9b1b422198944fbbd5250b3241691b6

Moreover, Esri/esri-loader is licensed under the Apache License 2.0. the Apache License 2.0 clearly stated that the information are freely available online, and any third party is permitted to access, download, copy, distribute, and use these materials in any way, even commercially, with proper attribution. More detailed contents can be viewed through the website:

https://github.com/Esri/esri-loader/blob/master/LICENSE

Based on the above information, we think Figure 1 can be published specifically under the CC BY 4.0 license.

Reviewer #1:

1. The paper is well written and developed

Response: Thank you for very much your positive comments. We appreciate your great input, which has helped us substantially improve the quality of our work. Please also see our point-by-point response to your comments.

2. The concept of socio-economic efficiency is not very clear from the abstract, please explain

Response: Investigation and assessment of ecological water resources is a question of rationalizing the use of resources and, especially, of reducing the use of scarce and limiting natural resources, or diminishing the use of other, potentially contaminating resources. In this sense, many studies have been dedicated to evaluating water use efficiency from a productive stance, and different economic or socio-economic water use indices with agricultural, industrial and ecosystem outputs have proposed such as water consumption per 10,000 yuan GDP or water consumption per capita. In this study, industrial and ecosystem outputs mainly refer to the development and utilization of the resources of salt lakes. 

As a decisions-making tool at economic level, socio-economic efficiency means using indicators of productivity, economic and even social to explore the utilization efficiency of water resources.

The references of the above contents are as follows：

Romero P, José García, Pablo Botía. 2005. Cost–benefit analysis of a regulated deficit-irrigated almond orchard under subsurface drip irrigation conditions in Southeastern Spain. Irrigation Science, 24(3):175-184.

Dichio B, Xiloyannis C, Sofo A, et al. 2007. Effects of post-harvest regulated deficit irrigation on carbohydrate and nitrogen partitioning, yield quality and vegetative growth of peach trees. Plant & Soil, 290(1-2):127-137.

Alkhamisi S. A., Abdelrahman H. A., Ahmed M., et al. 2011. Assessment of reclaimed water irrigation on growth, yield, and water-use efficiency of forage crops. Applied Water Science, 1(1):57-65.

José G G, Fulgencio Contreras L, Usai D, et al. 2013. Economic Assesment and Socio-Economic Evaluation of Water Use Efficiency in Artichoke Cultivation. Open Journal of Accounting, 2013, 2(2):45-52.

3. Socio-economic efficiency is not well discussed within the paper, please expand on this concept, what is being measured, what are the limitations of this concept and where has it been used before?

Response: The concept and application of socio-economic efficiency and the selective basis of indicators were added in page 13-14, line 293-313. The references were added to the list of references in page 29. The contents are as follows:

“As a decisions-making tool at economic level, socio-economic efficiency analysis uses productivity, economic and even certain social to explore the utilization efficiency of water resources [56]. The investigation and assessment of ecological water resources is a question of rationalizing the use of resources and, especially, of reducing the use of scarce and limited natural resources, or diminishing the use of other, potentially contaminating resources. In this sense, many studies have been dedicated to evaluating water use efficiency from a productive stance [57,58], and different economic or socio-economic water use indices with agricultural, industrial, and ecosystem outputs have been proposed, such as water consumption per 10,000 yuan GDP or water consumption per capita [59]. In previous research, most evaluations of the socio-economic efficiency of water resources targeted the management of an agricultural irrigation water system. The socio-economic efficiency of a water resources system is mainly influenced by the latter’s water resource use patterns and adjustment measures, which are controlled by human activities and the development of society and the economy [60]. Therefore, the selection of indicators is context-dependent and should be tailored to the actual situation of a given study area.

In our study, industrial and ecosystem outputs mainly refer to the development and utilization of salt lakes resources. 

Additionally, as one of the large drainage areas in the Qaidam Basin, the West Taijinar Lake is an important flood storage and detention area, yet also threatened by extreme flooding, which has certain effects upon local industrial production.”

At the same time, the discussion of the water consumption per 10,000 yuan GDP was enriched in page 21, line 463-467. The contents are as follows:

“From 2015 onward, the total industrial water consumption declined and total industrial output value rose yearly following the adjustments made to the production technology of Qinghai CITIC Guoan Lithium Development Co., Ltd. and Qinghai Hengxinrong Lithium Technology Co., Ltd. Consequently, the water consumption per 10,000 yuan GDP decreased on a yearly basis.”

The measurement of the flood control dam and socio-economic efficiency has been discussed in detail in page 21, line 469-475.

4. The motivation for this study area is not clear, I recommend the authors to emphasize why did the authors decide to examine this study area.

Response: In the abstract, I have explained the problems existing in the arid salt marsh area and the significance of this study in page 2, line 25-29 and line 47-49. At the same time, I elaborated the key role of water ecology in the exploitation of salt resources in the introduction in page 4, line 79-94. I also added the motivation for the study of West Taijinar Lake in page 4-5, line 95-108. The added contents are as follows:

“Currently, the Qaidam Basin Salt Lake is one of the largest inland salt lakes in China and harbors much industrial value. It has the longest exploitation history and the most mature technology to extract salt lake resources in China. With the extensive exploitation of its resources, the Qaidam Basin Salt Lake is facing several increasingly prominent problems, namely a drop in groundwater levels, a reduction in salt marsh areas, and an increase in desertification. Generally, when local overexploitation and local surplus co-occur, this adversely affects the comprehensive utilization of salt lake resources and their regional coordinated development. However, if water resources in this salt marsh area could be reasonably utilized to supplement the brine in the pressure-bearing layer, the sustainable exploitation of salt lake could be effectively realized. A case in point is the West Taijinar Lake, a seasonal tail lake in the middle of the Qaidam Basin. An investigation and assessment of ecological water resources in the salt marsh area of that lake has provided insight into the impact on water ecology from exploitation of salt lake resources. Such findings could prove useful for strengthening the rational utilization of water resources in salt marshes of those lakes and in other arid areas.”

5. I recommend authors to discuss the limitations found in the study.

Response: We have discussed the limitations found in the study in conclusion in page 22-23, line 500-504. The contents are as follows:

“The findings of this study must take into account some limitations. Firstly, due to time constraints, the surface water and groundwater throughout the Qaidam Basin could not be explored. Secondly, the proposed assessment indicator system for the ecological status of water resources is suitable for salt marshes in arid areas only; however, the index of the target layer can be added or subtracted according to the actual situation of a study area.”

6. I recommend authors to discuss future research avenues.

Response: The conclusion has raised the great challenges for the future of arid areas in page 23, line 504-507. The contents are as follows:

“Finding a way to accommodate and harness the uneven distribution of precipitation and flood could enable their full use for groundwater recharge, but this would be a major challenge to overcome in resolving the problems associated with falling groundwater levels and shrinking salt marshes in arid areas.”

Meanwhile, other future research avenues were discussed in conclusion in page 23, line 507-509:

“Based on the investigation and assessment of ecological water resources for the whole basin, it is imperative that future research avenues focus on the sustainable allocation of water resources, namely from rainfall and floods, in the Qaidam Basin.”

7. Please improve the language and flow of the paper for minor errors and typos.

Response: We have thoroughly copyedited our manuscript for language usage, spelling, and grammar with the help of Transcend Envirotech Consulting.

---

## [Editor Report · Decision Letter 1]

12 Jan 2021

Investigation and assessment of ecological water resources in the salt marsh area of a salt lake: A case study of West Taijinar Lake in the Qaidam Basin, China

PONE-D-20-31417R1

Dear Dr. Wang,

We’re pleased to inform you that your manuscript has been judged scientifically suitable for publication and will be formally accepted for publication once it meets all outstanding technical requirements.

Kind regards,

Ali Kharrazi

Academic Editor

PLOS ONE
---

## [Editor Report · Acceptance letter]

15 Jan 2021

PONE-D-20-31417R1 

Investigation and assessment of ecological water resources in the salt marsh area of a salt lake: A case study of West Taijinar Lake in the Qaidam Basin, China 

Dear Dr. Wang:

I'm pleased to inform you that your manuscript has been deemed suitable for publication in PLOS ONE. Congratulations! Your manuscript is now with our production department. 

Kind regards, 

on behalf of

Dr. Ali Kharrazi 

Academic Editor

PLOS ONE